# Multi-Objective Optimization of Liquid Silica Array Lenses Based on Latin Hypercube Sampling and Constrained Generative Inverse Design Networks

**DOI:** 10.3390/polym15030499

**Published:** 2023-01-18

**Authors:** Hanjui Chang, Shuzhou Lu, Yue Sun, Guangyi Zhang, Longshi Rao

**Affiliations:** 1Department of Mechanical Engineering, College of Engineering, Shantou University, Shantou 515063, China; 2Intelligent Manufacturing Key Laboratory of Ministry of Education, Shantou University, Shantou 515063, China

**Keywords:** liquid optical silicone lenses, residual stress, volume shrinkage, latin hypercube sampling, constrained generative inverse design networks, parameter optimization

## Abstract

**Highlights:**

**What are the main findings?
**
In this study, we apply the Latin hypercube sampling method for sampling and combine the CGIDN and response surface modeling methods, which can effectively optimize the injection process.The CGIDN method allows a small number of initial data points to be considered and uses a method of continuously updating the sampling points to guide the search for the optimal process parameters that minimize the residual stress values and volume shrinkage.

**What is the implication of the main finding?
**
The Latin hypercube sampling method allows uniform, random and orthogonal sampling within the planned spatial area of the experimental factor design and allows artificial control of the number of trials. The method yields sample data with high spatial coverage of the ex-perimental design can improve the accuracy of modeling.The method proposed in this study can effectively optimize the process parameters in the injection molding process, thus improving the reliability and quality output of the injection molded products and providing guidance to the plant engineers in adjusting the machines.

**Abstract:**

Injection molding process parameters have a great impact on plastic production quality, manufacturing cost, and molding efficiency. This study proposes to apply the method of Latin hypercube sampling, and to combine the response surface model and “Constraint Generation Inverse Design Network (CGIDN)” to achieve multi-objective optimization of the injection process, shorten the time to find the optimal process parameters, and improve the production efficiency of plastic parts. Taking the LSR lens array of automotive LED lights as the research object, the residual stress and volume shrinkage were taken as the optimization objectives, and the filling time, melt temperature, maturation time, and maturation pressure were taken as the influencing factors to obtain the optimization target values, and the response surface models between the volume shrinkage rate and the influencing factors were established. Based on the “Constraint-Generated Inverse Design Network”, the optimization was independently sought within the set parameters to obtain the optimal combination of process parameters to meet the injection molding quality of plastic parts. The results showed that the optimal residual stress value and volume shrinkage rate were 11.96 MPa and 4.88%, respectively, in the data set of 20 Latin test samples obtained based on Latin hypercube sampling, and the optimal residual stress value and volume shrinkage rate were 8.47 MPa and 2.83%, respectively, after optimization by the CGIDN method. The optimal process parameters obtained by CGIDN optimization were a melt temperature of 30 °C, filling time of 2.5 s, maturation pressure of 40 MPa, and maturation time of 15 s. The optimization results were obvious and showed the feasibility of the data-driven injection molding process optimization method based on the combination of Latin hypercube sampling and CGIDN.

## 1. Introduction 

In the auto lamp industry, the main considerations for choosing lens materials are light transmission, high-temperature resistance, aging resistance, life cycle, etc. The traditional glass lens has an excellent quality of high transmittance and high-temperature resistance, but has the disadvantage of high density, so the use of a glass lens will increase the weight of the lamp lens module, which is seriously inconsistent with the current development trend of the automotive industry toward lightweight parts. In addition, it is not easy to achieve large-scale production of the glass softened glass body of this particular state, there is a high cost of the mold, a long processing cycle, precision is not easy to control, and it is brittle, especially for the thickness of the relatively large changes in the lens, and accidental bursting can not be predicted or solved. Now this can be achieved by improving the process, only through the coating or toughening treatment to enhance the strength of the glass, but after these treatments, the glass lens light transmission will be reduced, which also increases the cost. So, at this stage plastic lenses are generally used instead of glass lenses. Commonly used optical grade plastic lenses for headlights generally use optical grade PMMA, optical grade PC, and other materials. The advantage of optical grade plastic is simple technology and low production costs, but the disadvantage is that high temperature and aging resistance is relatively low; PMMA temperature resistance is generally not more than 90 °C (heat deflection temperature 105 °C), PC temperature resistance is not more than 120 °C (heat deflection temperature 135 °C).

Compared with the traditional glass lens and plastic lens, the optical liquid silicone lens combines the advantages of the above-mentioned lenses. The new generation of optical liquid silicone lens temperature range is usually −40–250 °C, and it has the advantages of: resistance to sudden cold and heat shock, a flexible texture, being not easy to burst, excellent explosion-proof performance, non-toxicity, a low density, a stable performance, and a light transmission rate of up to 92%.

In early work, researchers attempted to improve plastic product quality by adjusting injection molding parameters and structural geometry. In 2017, a study by Chao-Ming Lin et al. [1] showed that Taguchi and CAE integrated methods can effectively optimize the optical properties of plastic Fresnel lenses, and it was shown that melt temperature, filling time, holding time, and mold temperature can affect the residual stresses within the Fresnel lens. In addition, setting optimal process parameters can optimize the lens quality and reduce the average residual stress by 75.1% and the relative average birefringence by 74.1%. In 2018, Mehdi Moayyedian et al. [2] optimized three common defects of injection molding, namely short shot, shrinkage, and warpage, and proposed a multi-objective optimization method based on the Taguchi method and fuzzy hierarchical analysis (FAHP). In 2017, Marcel Roeder et al. [3] found that three process parameters, melt temperature, mold temperature, and compression force, had a significant effect on the molding accuracy during the manufacturing of diffractive optical elements using injection compression molding. The experimental results showed that only precise control of each step of the molding process could produce complex polymer optical elements. However, these methods all share a common limitation, in that the optimal combination of process parameters is strongly dependent on the sampling interval or the level of parameters used in the DOE. In other words, the optimal result obtained by the DOE method may not be the global optimal solution.

The combination of sample points obtained based on DOE tests cannot cover the whole variable design space, which is prone to the problem of sample point stacking, and the accuracy of establishing an approximate model based on sample points is sometimes difficult to meet the analysis requirements; while uniform, random, orthogonal sampling within the design space region of the test factors obtained based on Latin hypercube sampling can obtain a large amount of model information with fewer data points.

Injection molding technology has many outstanding advantages: the ability to produce plastic parts with very small dimensions or with very complex surface structures, and a significant increase in production efficiency compared to grinding and polishing. To perfectly reproduce the high-precision microstructure of optical products, injection molding requires excellent mold design and processing and optimal process parameters. In the case of the basic replication rate of the shape, the imaging quality, and the optical index of the molded optical products, are not all satisfactory due to microstructure replication or internal stress, that is, the immaturity of the existing lens injection molding technology will further increase the cost of silicone lenses. Therefore, from the perspective of parameter optimization, the use of CAE analysis software and machine learning techniques to improve the injection molding quality of optical components can reduce the product scrap rate and improve economic efficiency, thus serving as a reference for actual production.

## 2. Literature and Review

In recent years, polymers have experienced tremendous growth in use in the production of optical components in special optical lenses such as, Fresnel lenses, free-form lenses, aspheric lenses, micro-lenses or micro-lens arrays, and diffractive optical elements. Optical polymers allow complex optical designs with multiple surfaces and various assembly features, and so far they have been widely used in various fields such as automotive [4], lighting [5], photovoltaic [6], electronics, and medical [7]. Among them, lens arrays are often used in compact and lightweight optical systems. Previously, lenses were made of glass. Nowadays, plastic materials (e.g., LSR, PMMA, PC, etc.) are used instead of glass to reduce manufacturing costs. The successful production of high-quality lens arrays requires not only good optical design, and precision mold-making, but also effective control of residual stresses during molding. Residual stresses have a great influence on the mechanical, thermal, and optical anisotropy of injection molded lenses, so an accurate analysis of them is essential. Although the effect of stress on the properties of molded parts is often discussed, the available methods and tools are not fully understood. Currently, there are two main types of evaluation of residual stresses in optical plastics: destructive tests and non-destructive tests.

In 2019, Yue, P. [8] analyzed the residual stress distribution in polycarbonate products during the molding process by birefringence experiments. The results showed that the birefringence in the thickness direction was influenced by the holding pressure and temperature. Annealing has little effect on birefringence and molecular orientation distribution. In 2009, Weng, C. et al. [9] evaluated the residual stresses after microchip forming using the birefringence method, and analyzed them using finite element numerical simulations. The experimental and simulated results were compared. The results showed that the most dominant process parameter was the temperature of the mold. As the mold temperature increased, the maximum residual stress value became smaller. In addition, micro-lens arrays are the core components of LED displays, which are generally produced using micro-injection molding technology. Due to the small size of micro-lens arrays, it is difficult to detect and characterize the residual stresses in the lens arrays. And the birefringence method remains a better method to measure the residual stresses in injection-molded optical parts. In 2007, Kim, C.H. et al. [10] used the incremental drilling method to determine residual stresses in polymer parts with complex geometries, and used the finite element method to determine the relaxation factor required for residual stress determination. The commercial software Moldflow predicts the residual stress distribution in polystyrene sheets. The results showed that the experimental results agreed with the predicted results. Thus, the incremental drilling method can be applied to the residual stress measurement of polymer materials, especially for the residual stress measurement of complex geometric parts.

The presence of residual stress directly affects the mechanical and optical properties of the part and can cause warpage and cracking in severe cases. In addition, it can greatly affect the replication quality of microstructures. Therefore, it is important to explore the molding mechanism and the influence mechanism of residual stress in the injection molding process for high-quality molding of polymers. Usually, high residual stresses cause optical distortion and deterioration of optical properties, so researchers usually measure and characterize residual stresses in lenses using birefringence techniques and incremental drilling methods and analyze residual stresses in polymer lenses using finite element numerical simulations and commercial software such as Moldflow.

Han-Jui Chang [11] used an identifiable method of photoelastic stress for measurement and compensation, combined with fuzzy theory, to reorganize a process that can be used to evaluate product re-residual stresses, resulting in an effective quantification and compensation measurement of residual stresses in products with the corresponding theoretical formulation. In 2015, Macías et al. [12] found that the quality of optical components made of transparent thermoplastic polymers depends on the presence of residual stresses, and that the wrong choice of process parameters can lead to the generation of residual stresses in plastic lenses, while residual stresses in the components can significantly affect the structural dimensions of the lenses and their dimensional accuracy. In 2021, Wei-T ai Huang et al. [13] optimized the design for the influencing factors such as injection time, base material temperature, mold temperature, injection pressure, holding pressure, holding time, coolant, and cooling temperature. Warpage and temperature distribution were analyzed as performance indicators. Then the signal-to-noise ratio (S/N ratio) was calculated. Based on the Gray correlation analysis, multiple performance characteristics indexes were obtained and the maximum multiple performance characteristics index was used to find out the multiple quality characteristics to optimize the process parameters. In 2022, Xiaoyu Zhang et al. [14] used molecular dynamics software to develop a simulation model for injection molding of micro-pillar arrays, and showed that increasing the crystallizer temperature and melt temperature would reduce the thermal residual stress and molecular orientation stress, which would result in a more uniform distribution of residual stresses, while on the contrary, increasing the packing pressure would result in a stronger flow shear field and increase the molecular orientation. The increase of packing pressure will make the flow shear field stronger and increase the molecular orientation stress, which will further aggravate the residual stress. In 2018, Salmoria, G.V. et al. [15] used morphological and mechanical properties of poly(L-co-D, l -lactic acid) (PLDLA) specimens injection molded at different melt temperatures and stress concentrators. The results showed that at lower melt injection temperatures, the specimens underwent birefringence along the surface, i.e., the presence of residual stresses due to the filling phase and rapid solidification. On the other hand, high-temperature injected specimens showed residual stress concentrations near the gate due to the filling effect of the packing pressure.

Optical lenses have high-quality requirements, and the difficulty lies in the precise control of lens geometric precision, optical properties, and molding precision of optical surface microstructure. Due to the inherent characteristics of polymer materials, such as large thermal expansion and contraction effects, the existence of molecular orientation, birefringence phenomenon, etc., it is difficult to meet the high-quality requirements of optical lenses by adopting an ordinary injection molding process. And the residual stress will affect the physical properties of reproduction quality, size, and microstructure. Therefore, studying the influence of process parameters on the residual stress of injection molded parts can be of great help in the production of micro structured parts. The results of the above study show that melt temperature, mold temperature, and holding pressure are important factors affecting the residual stresses within the lens.

With the wide application of computer-aided engineering, there are more and more optimization methods based on fitting techniques. In 2021, Jinsu Gim et al. [16] proposed a method to analyze the effect of cavity pressure distribution on part quality using neural networks. The process state points extracted from the cavity pressure profile were used as input features to the model, while the relationship between the cavity pressure distribution and the part weight was output as a quality indicator. The influencing features and impacts were analyzed to allow the target points and boundaries of the monitoring window to be determined, and the contribution of each feature was used to optimize the injection molding process. In 2022, Han-Jui Chang et al. [17] proposed a non-explicit genetic algorithm for the multi-objective optimal design of UAV shells, in which process parameters such as melt temperature, filling time, pressure, and pressure time were investigated as model variables, and kriging response surface analysis was used to analyze the sampled point data to obtain the warpage values, die stamping indices, and mathematical relationship, and then the multi-objective optimization procedure of the genetic algorithm was used instead of analyzing the experiments. The results showed that the optimization rate of the die index reached 96.2% through the optimization nodes of the genetic algorithm and the experimental verification, and the average optimization rate of the four main optimization nodes was 91.2%, and the error rate with the actual situation was only 8.48%, which met the needs of the actual production.

In summary, many scholars have used artificial intelligence techniques to predict the quality of injection molded parts, achieving intelligent and automated results [18,19]. As shown in Figure 1, this study conducts experiments on four key factors, namely filling time, melt temperature, ripening time, and ripening pressure, to reduce the residual stress value of the lens array to obtain the optimal process parameter settings and improve the optical performance of the lens array. In addition, in order to predict the residual stress and volume shrinkage in the molded lens and save the time cost of the experiment, this study proposes a CGIDN-based, data-driven injection molding process optimization framework based on the data obtained from the experiment based on the Python platform, which can obtain the optimal process parameters for the lens residual stress value and volume shrinkage up to the minimum.

## 3. Methodology

### 3.1. Latin Hypercube Sampling (LSH)

Latin hypercube sampling was proposed by McKay et al. [20] in 1979. However, the same sampling technique was independently proposed by Eglājs in 1977, and further developed by Ronald L. Iman et al. [21] in 1981. Latin hypercube sampling (LHS) is a method of approximate random sampling from a multivariate parameter distribution, which is a stratified sampling technique, often used in computer experiments or Monte Carlo integration, etc.

Latin hypercube sampling (LHS), a stratified random sampling, can sample efficiently from the distribution interval of variables. Assuming that there are now *k* variables $x_{1},x_{2},$ ……. $x_{k-1},x_{k}$, and we now want to take N samples from their specified intervals, then the cumulative distribution of each variable is divided into the same N small intervals, and a value is randomly selected from each interval, and the N values of each variable are randomly combined with the values of the other variables. Unlike random sampling, this method is able to ensure full coverage of the range of variables by maximizing the stratification of each marginal distribution.

In injection molding multi-objective optimization, most of the experimental analysis samples are obtained by orthogonal tests. The combinations of sample points obtained based on orthogonal tests sometimes do not fill the entire design space of the variables and are prone to the problem of sample point accumulation, and the accuracy of the approximate model based on orthogonal tests is sometimes difficult to meet the analysis requirements.

The Latin hypercube sampling method is a randomized multi-dimensional stratified sampling method, in which the probability distribution function of the test factors is divided into N non-overlapping subregions according to the value range of the test factors, and then independent equal-probability sampling is performed in each subregion. Compared with the orthogonal test, the Latin hypercubic design is more relaxed in the hierarchy of level values, and the number of trials can be controlled artificially. However, the distribution of test points may not be uniform enough, and the possibility of losing some areas of the design space increases as the number of levels increases. The optimal Latin hypercube sampling method improves on the Latin hypercube sampling method by sampling uniformly, randomly, and orthogonally over the design space of the test factors, which can obtain a large amount of model information with a relatively small number of points.

### 3.2. Constrained Generative Inverse Design Networks

GIDN (Generative Inverse Design Networks) uses backpropagation with resolved gradients, allowing for fast computation on a variety of inputs while avoiding getting stuck in local optima. In addition, the amount of data required can be reduced as GIDN is combined with active learning to reach the optimal solution step by step. The flow of the GIDN method is as follows: firstly, as with traditional deep neural network (DNN) training, the weights and deviations of the DNN are trained based on the relationship between the inputs and outputs. Secondly, the weights and deviations of the trained DNN are considered as fixed constants, and the input parameters of the minimum objective function are obtained by backpropagation.

The output values of the recommended inputs are calculated by simulation or experiment, and the newly generated data is combined with the previous data to update the neural network. These processes are repeated until the optimum value is reached. A detailed explanation of the GIDN method can be found in a previous study by Chen & Gu [22]. Unfortunately, however, the original GIDN method has the disadvantage that the range of input parameters is unbounded. The CGIDN method is depicted in Figure 2 and consists of two DNNs: a “predictor” and a “designer”. Both DNNs have the same neural network structure. Hyperparameters, including the number of hidden layers and neurons, are tuned to balance prediction accuracy and computational cost. (Figure 2 shows the structure of the network containing hyperparameters, where the number of hidden layers and neurons will directly affect the actual performance of the model. Table 1 shows the initial inputs for training the CGIDN. And Figure 6 shows the iterative results of CGIDN.) The predictor is a forward prediction model that is trained to approximate a physics-based model (or an arbitrary function). The learning variables in the predictor are the weights and biases of the neurons connecting adjacent layers. After training, the weights and deviation values in the predictor are assigned to the designer.

As shown in Figure 3, the main steps of this study are roughly divided into two stages: the first stage is to select the process parameters affecting lens forming (such as filling time, melt temperature, maturation time, and maturation pressure), set up a reasonable combination of tests and number of tests, according to the Latin hypercube sampling method, build a response surface model according to the experimental results, and finally confirm the best process parameters according to the response surface model. The second stage is to divide the test results obtained in the first stage into a training set and a test set to build and train the CGIDN model, and finally to use the trained model to search for the best process parameters. This search process can be represented as the initial design, with Gaussian distribution values as the initial values, which are input into the designer. The output of the optimal design is then generated based on the resolved gradient calculated by backpropagation. In a feedback loop, the optimized design is validated by the physically based model, and can be added to the previous training data for the next iteration of the training and design process until the desired optimized properties are maximized (or minimized) and then stopped.

## 4. Case Study

In this study, we take the produced automotive liquid optical silicone lens array as the object of study. The automotive liquid optical silicone lens array as shown in Figure 4, the maximum flesh thickness is about 22,483 mm, the minimum flesh thickness is about 0.101 mm, and the volume is 53,972.35$\;{\mathrm{mm}}^{3}$. After the dimensional measurement of the automotive liquid optical silicone lens array, the modeling of the lens array is carried out using the SolidWorks software to model the lens array. Experiments were conducted using the LSR material, model LSR-1, and from CAE, and the PVT curve of this material is shown in Figure 5. In addition, experiments were conducted on an all-electric injection molding machine (KM 50–250 PX, Krauss Maffei, Room 623, 6F, No.88 Taigu Road, China (Shanghai) Pilot Free Trade Zone, Shanghai, China), while the prediction model was built on a Python 3.10 platform.

Compared to PMMA and PC, liquid silicone rubber (LSR) has many advantages due to its heat and weather resistance, non-toxicity, electrical insulation, biocompatibility, non-color transparency, tear-resistance, etc., and is widely used in medical, electronic and electrical, food, and mother and baby industries. LSR is an ideal material for very demanding applications due to its unique properties. In the field of electronic packaging, LSR is mainly used for the packaging of electronic products, which can be sealed, water-resistant, dust-resistant, heat-conducting, shock-resistant, and insulated. In addition, LSR is often used in the production of plastic lenses for LED lamps because of its high transparency and high refractive index.

Optical lenses have high quality requirements, and the difficulty lies in the precise control of lens geometric precision, optical properties, and molding precision of optical surface microstructure, reducing processing costs and improving production efficiency. Due to the inherent characteristics of polymer materials, such as thermal expansion and contraction, molecular orientation, and birefringence, it is difficult to meet the high-quality requirements of optical lenses by the common injection molding process. In addition, the microstructures on the surface of polymer optical lenses are on the scale of microns or even nanometers, and the uneven wall thickness of optical plastic lenses makes the density and refractive index of the plastic lenses molded by common injection molding methods unevenly distributed, and the lenses produce residual internal stress and birefringence, and the surface shape accuracy cannot meet the requirements of use. The ultimate goal of this study is to adjust reasonable process parameters to improve the quality and yield of the lens.

## 5. Discussion

To determine the key control parameters affecting the optical properties of the lenses, the filling time (A), melt temperature (B), maturation pressure (C) and maturation time (D) were used as test factors, and the volume shrinkage (F) and residual stress values (G) of the silicone lens arrays were used as optimization targets. The recommended range of levels for the four test factors is shown in Table 1. 20. Latin test samples were obtained for the four test factors based on the Latin hypercube sampling method, and the results of the optimization objectives were obtained by testing as shown in Table 2, with the mean values of residual stress and volume shrinkage shown in columns 6 and 7 respectively. In contrast to the DOE test, the samples obtained from the Latin hypercube test are not limited to the levels within the parameter settings, and the number of samples set can be controlled artificially.

The optical properties of injection molded optical products are mainly influenced by the birefringence properties and surface geometry. According to the photometric elastography theory, the birefringence of isotropic polymer chains is mainly related to residual stresses. In the case of injection molded lenses, for example, the residual stresses come from two parts of the molding fundamentals: those caused by the filling flow and those caused by the thermal change process. In this paper, among the 20 combinations of experiments obtained based on the LSH method, sampling as the evaluation index of residual stress and volume shrinkage of automotive lamp lens arrays, the most favorable combination for the average residual stress and volume shrinkage is experiment 14. The optimal process parameters are melt temperature of 24,524 °C, filling time of 2.4841 s, curing time of 29.25 s. The residual stress value in the lens was 11,963 MPa and the volume shrinkage was 4.883% under the process parameters of 25,833 MPa.

CGIDN was applied to optimize the injection molding process of the lens. Twenty randomly generated initial data points were considered. As shown in Figure 6, the CGIDN results were plotted for every 30 iterations of the optimization cycle, forming the Pareto front curve as optimized. Although the optimal process parameters were reached in the 88th cycle, 150 optimization cycles were executed to verify convergence. The obtained process parameters minimized both the residual stress values and the volume shrinkage, as shown in Figure 6. The point with the minimum volume shrinkage was selected as the optimal point of the process among the points with the minimum residual stress. The optimized residual stress and volume shrinkage were 8.47 MPa and 2.83%, respectively. Compared with the initial data set, the optimized process conditions improved 29.18% and 42% in residual stress value and volume shrinkage, respectively, as shown in Table 3. The optimal process parameters optimized by the CGIDN method were melt temperature of 30 °C, filling time of 2.5 s, maturation pressure of 40 MPa, and maturation time of 15 s.

**Figure 6 polymers-15-00499-f006:**
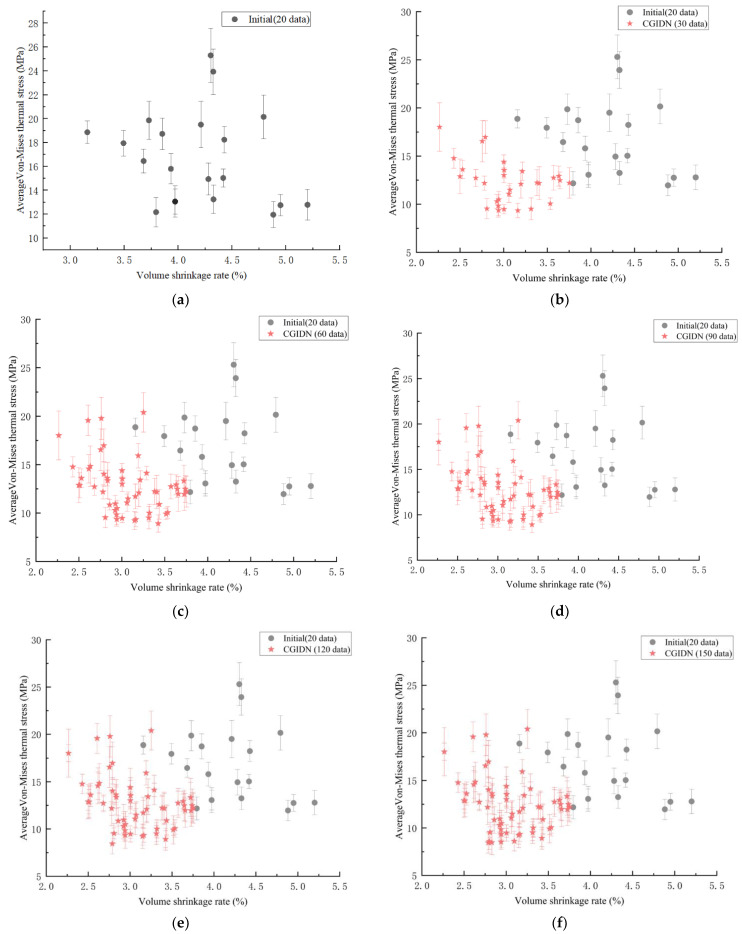
The results of the CGIDN are plotted at every 30 iterations of the operation optimization loop. (**a**) 20 initial data points for random search; (**b**) 20 initial data points and 30 iteration points (CGIDN); (**c**) 20 initial data points and 60 iteration points (CGIDN); (**d**) 20 initial data points and 90 iteration points (CGIDN); (**e**) 20 initial data points and 120 iteration points (CGIDN); (**f**) 20 initial data points and 150 iteration points (CGIDN).

Figure 7 shows the residual stress index points of the 10 lens surfaces before optimization. The smaller the residual stress value, the smaller the lens birefringence, so they represent the optical quality of the lens. The green area represents a larger index position and a greater birefringence, and the blue area represents a smaller index, indicating a better optical performance at that position. The residual stress values of the 10 test points before optimization were 23.976, 24.416, 22.497, 17.744, 22.574, 27.695, 18.489, 23.627, 19.784, 29.423 MPa, and the average residual stress value before optimization was 23.0225 MPa. Figure 8 shows the residual stress index points of the 10 optimized lens surfaces. The residual stress values of the 10 optimized test points are 26.390, 15.869, 18.434, 15.226, 13.548, 14.781, 14.125, 17.275, 15.691, 21.744 MPa, and the average residual stress value after optimization is 17.3083 MPa. The overall reduction of residual stress on the lens surface is about 5.7142 MPa, which is 24.8% of the optimization rate.

The injection molding industry is a high energy consumption and large carbon emission industry. According to statistics, the annual energy consumption of the global injection molding industry is about 30 billion kilowatt hours, accounting for about 10% of the total global energy consumption, and carbon emissions are also quite alarming. Among them, the injection molding machine is the major energy consumer; its energy consumption per unit of product is four times that of ordinary electrical equipment, if added with other auxiliary systems, energy consumption will be even greater. There are two ways to improve energy saving and emission reduction: one is to improve the injection molding machine, and the other is to use better quality materials. For example, try to use a low energy consumption, high efficiency injection molding machine. Use a high precision servo motor, ball screw, lubrication system, pressure sensor, and insulation cover to keep the motor performance stable all year round. Compared with hydraulic machines, the consumption of electric and water energy can be greatly reduced. Usually, the injection temperature of PC material is 250 °C to 330 °C, the injection temperature of PMMA material is 220 °C to 270 °C, while the injection temperature of LSR material is between 10 °C to 30 °C. The use of LSR material can greatly reduce the power consumed by the injection molding machine to heat the plastic, and realize the energy-saving and low-carbon production process. In addition, due to the characteristics of LSR material, it is not easy to warp the injection molded parts, so if the volume change of the products before and after V/P switching is too large, it is necessary to increase the holding pressure and extend the holding time, to increase the compensatory shrinkage, but this causes an increase in carbon emissions, so the volume shrinkage rate can indirectly represent the amount of carbon emissions, equivalent to the injection molding fingerprint. If the parameters of the process change are not adjusted properly, there will be greater volume shrinkage, and carbon emissions will be increased. From the test results in Table 2, it is seen that the maximum value of volume shrinkage is 5.200% and the minimum value is 3.157% in 20 sets of tests, which is a difference of 2.043%. For the all-weather, large-scale and high-production injection molding industry, a 1% reduction in volume shrinkage per part is equivalent to a reduction in carbon emissions of hundreds of thousands of tons per year. Therefore, in order to achieve the goal of “carbon neutrality”, we need to control the injection molding process parameters.

Therefore, we observe the influence between factors on the target by response surface plots in order to analyze the interaction between factors more intuitively. Figure 9 shows the three-dimensional surface of the response model, reflecting the effect of the interaction of the four factors on the volumetric shrinkage of the plastic part. Figure 9a shows the melt temperature and filling time_volume shrinkage response surface graphs. From Figure 9a, the volume shrinkage of the lens is minimized when the melt temperature is between 25 and 30 °C and the filling time is between 2.2 and 2.4 s, and the volume shrinkage value is around 3%. Figure 9b shows the graph of melt temperature and curing time_volume shrinkage response surface. From Figure 9b, the volume shrinkage of the lens is minimized when the melt temperature is between 25 and 29 °C and the curing time is between 14 and 18 s, and the volume shrinkage value remains below 3%. Figure 9c shows the response surface graphs of melt temperature and curing pressure_volume shrinkage. From Figure 9c, it can be seen that the volume shrinkage of the lens gradually decreases from 5% to less than 3% as the curing pressure increases when the melt temperature is within 20 to 30 °C. Figure 9d shows the response surface graph of curing pressure and curing time_volume shrinkage. From Figure 9d, it can be seen that the volume shrinkage of the lens decreases as the curing pressure increases and the curing time decreases, and the volume shrinkage reaches the minimum value of 2.9% when the curing pressure is 40.66 MPa and the curing time is 14.37 s. Figure 9e shows the response surface graph of curing pressure and filling time_volume shrinkage rate. From Figure 9e, it can be seen that the volume shrinkage of the lens is smaller with a value of 3% or less when the filling time is 2.0 s to 2.2 s and the curing pressure is higher. Figure 9f shows the response surface graphs of curing pressure and filling time_volume shrinkage. From Figure 9f, it can be seen that when the filling time is 2.0 s to 2.2 s, the smaller the curing time is, the smaller the volume shrinkage of the lens is, and the value is around 2.5%.

Normally, the shrinkage of the product in the holding and cooling stage should increase with the increase of the melt temperature, but LSR injection molding gets just the opposite conclusion, which can be explained by the low viscosity of LSR. The melt viscosity will become smaller after the increase of the material’s temperature, if the injection pressure and curing pressure remain unchanged at this time, the the freezing speed of the gate will be slowed down, which means that the holding time will be extended and the shrinkage will be increased. The shrinkage rate decreases as the density increases. In addition, a higher curing pressure and lower curing time can make the product in the cavity dense and reduce shrinkage, especially the pressure in the curing stage has more influence on the shrinkage of the product. This can be explained by the fact that the molten resin is compressed under the action of molding pressure, and the higher the pressure, the greater the compression that occurs, and the greater the elastic recovery after the pressure is lifted, making the plastic part size closer to the cavity size, and therefore the shrinkage less. In summary, the volume shrinkage of an LSR lens array reaches the minimum value (less than 3%) when the filling time is 2.2 to 2.4 s, the melt temperature is 25 to 30 °C, the curing pressure is close to 40 MPa, and the holding time is close to 15 s, which means that the carbon emission is minimal under these parameters.

## 6. Conclusions

Injection molding is one of the most important plastic molding methods today, and the setting of its process parameters directly affects the molding quality of plastic parts, so it is necessary to improve the molding quality of injection molded parts by optimizing the process parameters of injection molded parts. Although process optimization is important to achieve high quality at low cost, field engineers usually find the process conditions by heuristic methods. Therefore, in this paper, we take an automotive lamp lens array as an example and use SolidWorks to build a product model of an automotive lamp lens array. Based on the Latin hypercube sampling method, we obtain 20 sets of Latin test samples and train the test data obtained from injection molding tests using machine learning algorithms in artificial intelligence technology. and volume shrinkage to improve the quality of the lens. These studies provide theoretical guidance and practical reference for improving the molding quality of automotive plastic lenses. The main conclusions obtained from the study findings are as follows.
For the four parameters of melt temperature, filling time, maturation time and maturation pressure, 20 sets of Latin test samples were obtained based on the Latin hypercube sampling method with the range of test factor levels in Table 1, and the results of the optimization objectives were obtained through the tests. The optimal process parameters were melt temperature of 24.52 °C, filling time of 2.48 s, maturation time of 29.25 s and maturation pressure of 25.83 MPa, under which the residual stress value in the lens was 11.96 MPa and the volume shrinkage was 4.88%.The tests showed the relationship between the effect of the four process parameters on the volume shrinkage and the average residual stress. The influencing factor is mainly the melt temperature, followed by the curing time and curing pressure, and finally the filling time. Therefore, the four influential factors are ranked as follows: melt temperature > curing time > curing pressure > filling time. In addition, the average residual stress value on the surface of the lens before optimization was 23.0225 MPa, while the average residual stress value after optimization was 17.3083 MPa. The overall reduction of residual stress on the lens surface was about 5.7142 MPa, and the optimization rate was 24.8%.Parameter optimization for injection molding is equivalent to a black-box optimization problem, and CGIDN is a good choice for such problems, especially when rich data are difficult to obtain due to high computational cost or time. In this study, CGIDN is applied to optimize the injection molding process of lenses. Twenty randomly generated initial data points were considered. The CGIDN results were plotted at every 30 iterations of the optimization cycle, and although the optimal process parameters were reached in the 88th cycle, 150 optimization cycles were executed to verify convergence. The obtained process parameters simultaneously minimize the residual stress values. Among the points where the residual stresses were minimized, the point with the lowest volume shrinkage was selected as the best point for the process. The optimized residual stresses and volume shrinkage were 8.47 MPa and 2.83%, respectively. Compared with the initial data set, the optimized process conditions showed an improvement of 29.18% and 42% in residual stress values and volume shrinkage, respectively. The optimal process parameters optimized by the CGIDN method were a melt temperature of 30 °C, a filling time of 2.5 s, a maturation pressure of 40 MPa, and a maturation time of 15 s.Higher curing pressure and lower curing time can make the product in the cavity dense, shrinkage is reduced, especially the pressure of the curing stage has a greater impact on the shrinkage rate of the product. This can be explained by the fact that the molten resin is compressed under the action of the molding pressure, and the higher the pressure, the greater the compression that occurs, and the greater the elastic recovery after the pressure is released, which makes the size of the molded part closer to the cavity size, and therefore shrinkage is smaller. In this study, the volume shrinkage of LSR lens arrays reached the minimum value (less than 3%) when the filling time was 2.2 to 2.4 s, the melt temperature was 25 to 30 °C, the curing pressure was close to 40 MPa, and the holding time was close to 15 s, which means that the carbon emission was minimal under these parameters.

## Figures and Tables

**Figure 1 polymers-15-00499-f001:**
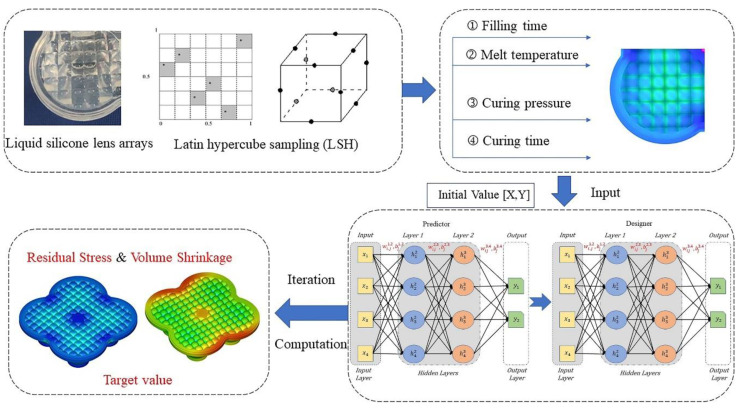
CGIDN Optimization Concept for optical components experimental.

**Figure 2 polymers-15-00499-f002:**
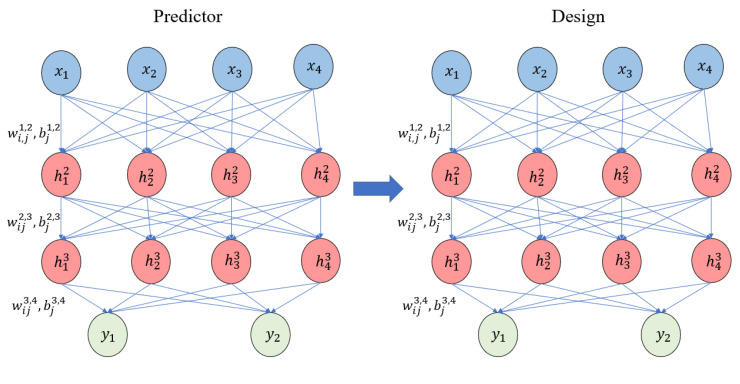
The framework of Generative Inverse Design Networks (GIDNs). GIDNs consist of two DNNs: the predictor and the designer.

**Figure 3 polymers-15-00499-f003:**
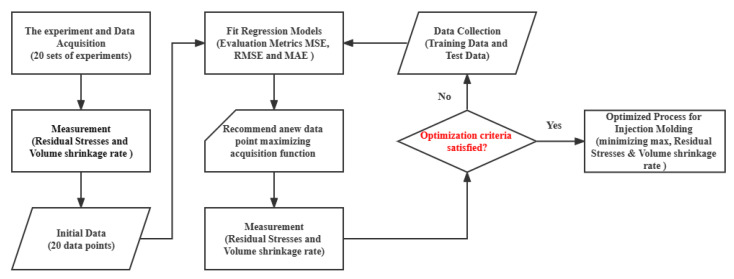
Flowchart of forecasting model modeling.

**Figure 4 polymers-15-00499-f004:**
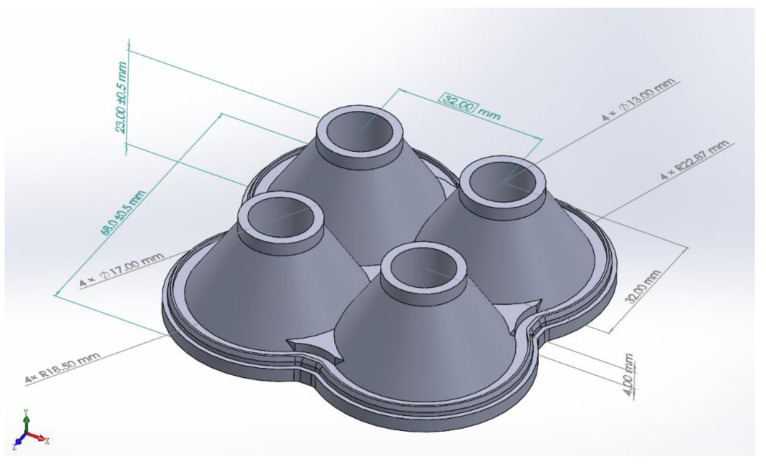
Physical view of the optical silicone lens array.

**Figure 5 polymers-15-00499-f005:**
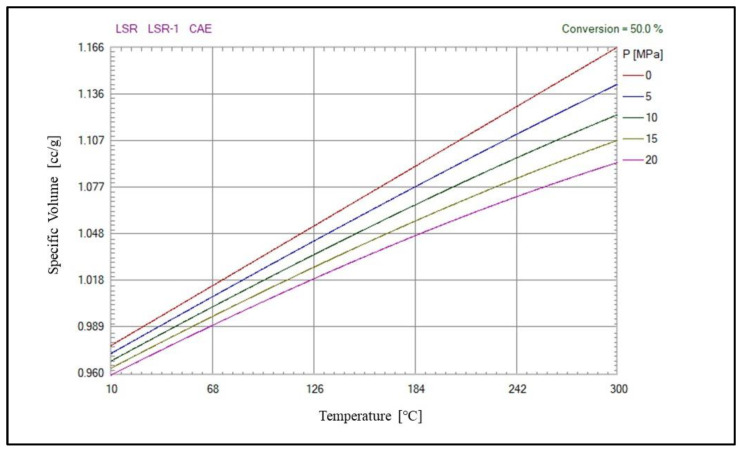
PVT curves of the original LSR material in this study.

**Figure 7 polymers-15-00499-f007:**
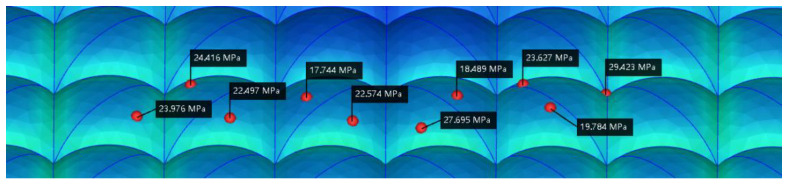
Graph of residual stress data on the lens surface before optimization.

**Figure 8 polymers-15-00499-f008:**
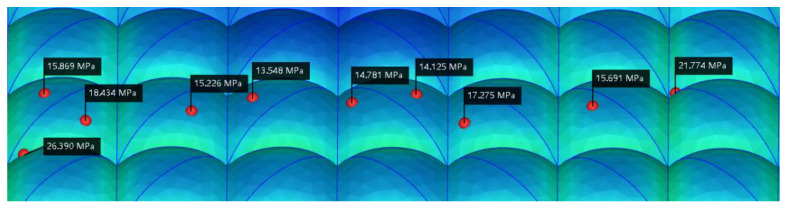
Graph of residual stress data on lens surface after optimization.

**Figure 9 polymers-15-00499-f009:**
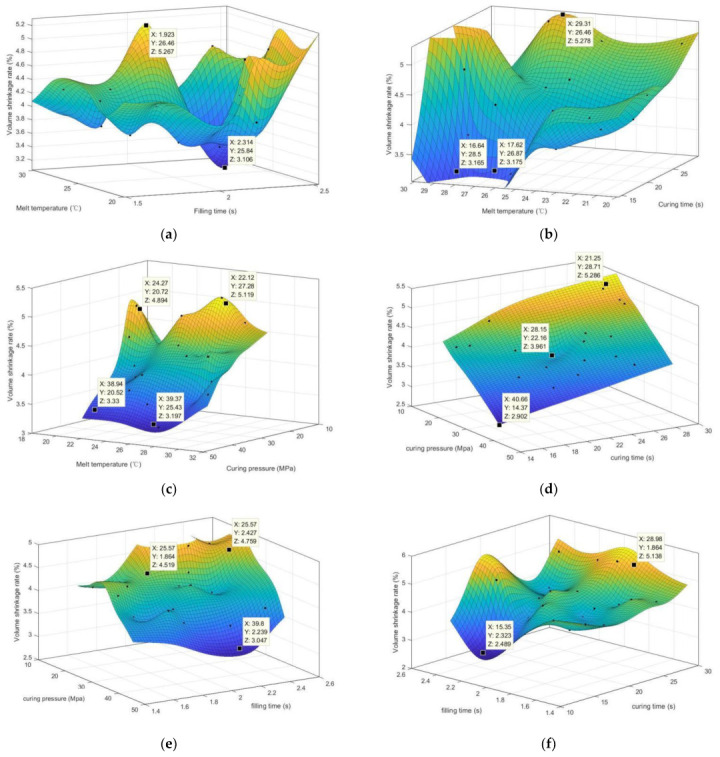
Response surface graph of volumetric shrinkage. (**a**) the response surface graphs of melt temperature and filling time_volume shrinkage; (**b**) the response surface graphs of melt temperature and curing time_volume shrinkage; (**c**) the response surface graphs of melt temperature and curing pressure_volume shrinkage; (**d**) the response surface graphs of curing pressure and curing time_volume shrinkage; (**e**) the response surface graphs of curing pressure and filling time_volume shrinkage; (**f**) the response surface graphs of filling time and curing time_volume shrinkage.

**Table 1 polymers-15-00499-t001:** Average Von-Mises thermal stress changes with melt temperature.

Factor	Description (Unit)	Minimum Value	Maximum Value
A	Filling time (s)	1.5	2.5
B	Melt temperature (°C)	20	30
D	Curing pressure (MPa)	20	40
E	Curing time (s)	15	30

**Table 2 polymers-15-00499-t002:** Results of 20 sets of Latin test samples and optimization targets.

Group	Filling Time (s)	Melt Temperature (°C)	Curing Pressure (MPa)	Curing Time (s)	Average Von-Mises Thermal Stress (Mpa)	Volume Shrinkage Rate (%)
1	1.82	29.31	35.64	26.08	13.06	3.97
2	2.38	28.53	20.90	18.30	20.16	4.79
3	1.56	27.99	31.35	26.67	13.25	4.33
4	2.07	26.06	30.64	24.45	14.95	4.28
5	1.77	28.26	33.08	17.97	19.86	3.73
6	2.13	20.37	26.30	21.70	18.24	4.43
7	2.04	27.15	27.85	18.93	19.51	4.21
8	2.34	25.94	39.72	17.24	18.87	3.16
9	1.66	25.02	29.04	24.98	15.04	4.42
10	1.96	26.89	21.97	28.26	12.79	5.20
11	2.22	24.45	37.32	20.06	17.95	3.49
12	1.74	21.51	22.06	15.08	25.30	4.30
13	1.64	22.37	32.56	20.58	18.73	3.85
14	2.48	24.52	25.83	29.25	11.96	4.88
15	1.54	23.58	23.01	16.05	23.94	4.33
16	1.92	22.74	36.29	22.51	16.45	3.68
17	2.16	20.52	24.55	29.13	12.76	4.95
18	1.89	23.24	34.02	23.65	15.81	3.93
19	2.44	29.68	38.38	27.07	12.18	3.80
20	2.26	21.21	28.06	22.35	13.06	3.97
Mean	2.00	24.97	29.93	22.52	16.69	4.19
SD	0.290	2.903	5.714	4.305	3.793	0.503

**Table 3 polymers-15-00499-t003:** Performance comparison between optimal parameters from CGIDN and best initial parameters.

Optimization Objectives	Best among Initial	Optimum by CGIDN	Improvement (%)
**AverageVon-Mises thermal stress**	11.96 MPa	8.47 MPa	29.18
**Volume shrinkage rate**	4.88%	2.83 %	42

## Data Availability

Not applicable.

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
