# Peer review of "Multi-Objective Optimization of Liquid Silica Array Lenses Based on Latin Hypercube Sampling and Constrained Generative Inverse Design Networks"

_polymers, 2023, doi:10.3390/polym15030499_

Round 1

Reviewer 1 Report

This work is not focused on Polymer science and therefore I think it is not appropriate for this journal. On the other hand this is an editorial choice and my tecnical comments are provided below.

- The paragraphs/structure of the MS is not typical for Polymer journal. The Highlights are long and the “literature” section is not needed as the state-of-art should be included in the introduction.

- Introduction is divided in subparagraphs and each reports a summary of a paper. It should be rewritten and be a critical analysis of the literature in the domain.

- Figure 2 is already included in figure 1.

- What about statistical errors for parameters in Table 2?

Author Response

Dear Sir,   Appreciate for your valuable comments, it's help our team going to much better, please allowed me to say thanks to you again.   Best Regards, Han-Jui Chang

Reviewer 2 Report

The paper is focused on the multiobjective optimization of Liquid Silica Array Lenses. The topic falls within the scope of the journal. I recommend the publication after the following minor revisions:

-          Paragraph 2 should be shortened being that the paper is a research article (it is not a review).

-          Table 2. Please add the errors for all results.

-          Error bars should be added in Fig. 6.

Author Response

(The authors gave the same response as above.)

Reviewer 3 Report

The work entitled: "Multi-objective Optimization of Liquid Silica Array Lenses Based on Latin Hypercube Sampling and Constrained Generative Inverse Design Networks" is a nice work exhibiting the successful combination of the modeling and polymer processing for the fabrication of transparent polymer specimens for the lenses and other applications in industry.

I would recommend the manuscript for publication after some modifications to the following points:

- Figure 1: The Target Value should be depicted more precisely in the image.

- Is really necessary the accuracy of the third digit in the performance comparison and also in the characterization results? It would be better to have an average number and an estimated statistical error.

- The phrase: "Injection molding machine plays an important role in injection molding, and there are two ways to improve energy saving and emission reduction: one is to improve from the injection molding machine, and the other is to use better quality materials" is very generic. Please be specific. A new machine will have better energy consumption parts, and a better quality material will have faster processing time, less failure etc...

- I do agree that polymer processing is of great importance, but many components are combinations of different polymers and also different materials (e.g., polymer and inorganic particles). I am wondering, can the authors make a statement on the viability of the model for composites processing?

Comment: I like the highlight part at the beginning of the paper. However, I am wondering if it fits the journal. I recommend including this in the conclusion and fusing it with the points you have there.

Author Response

(The authors gave the same response as above.)

Round 2

Reviewer 1 Report

all comments addressed

Reviewer 3 Report

THe manuscript can now get accepted for publication in Polymers.